# Valeric Acid: A Gut-Derived Metabolite as a Potential Epigenetic Modulator of Neuroinflammation in the Gut–Brain Axis

**DOI:** 10.3390/cells14221823

**Published:** 2025-11-20

**Authors:** Chiara Paciolla, Michele Manganelli, Mariagiovanna Di Chiano, Francesca Montenegro, Anna Gallone, Fabio Sallustio, Gabriella Guida

**Affiliations:** 1Department of Traslational Biomedicine and Neuroscience (DiBraiN), University of Bari Aldo Moro, 70121 Bari, Italy; c.paciolla3@phd.uniba.it (C.P.); m.manganelli1991@gmail.com (M.M.); mariagiovanna.dichiano@uniba.it (M.D.C.); anna.gallone@uniba.it (A.G.); 2Department of Precision and Regenerative Medicine and Ionian Area (DiMePRe-J), University of Bari Aldo Moro, 70121 Bari, Italy; francesca.montenegro@uniba.it (F.M.); fabio.sallustio@uniba.it (F.S.)

**Keywords:** microbiota, SCFA, neuroinflammation, valeric acid, HDAC

## Abstract

**Highlights:**

**What are the main findings?**
Valeric Acid (VA), a gut-derived short-chain fatty acid (SCFA), acts as a selective inhibitor of Class I Histone Deacetylases (HDACs), particularly HDAC3.VA modulates neuroinflammation and promotes neuroprotection by both epigenetic and GABAergic mechanisms.

**What are the implications of the main findings?**
VA offers a safer, physiological strategy to the non-selective pharmacological analogue, Valproic Acid (VPA), which is limited by significant systemic toxicity.Elucidating VA’s role might promote microbiome-derived compounds for targeted epigenetic modulation of neurodegenerative disorders.

**Abstract:**

The gut–brain axis (GBA) is a critical area of research for understanding the pathogenesis of neuroinflammatory and neurodegenerative diseases. Metabolites produced by the gut microbiota, particularly short-chain fatty acids (SCFAs), act as key mediators in this bidirectional communication. While the roles of acetate, propionate, and butyrate are well-established, valeric acid (VA), a five-carbon SCFA, is poorly understood. This comprehensive review explores VA as a gut-derived physiological epigenetic modulator, examining its microbial biosynthesis and systemic effects. This review discusses how VA acts as a selective histone deacetylase inhibitor (HDACi), particularly targeting Class I HDACs, to modulate gene expression and exert neuroprotective and anti-inflammatory effects. The analysis compares VA with its pharmacological analog, valproic acid (VPA), a well-known but non-selective HDACi. This comparison highlights how VA’s physiological nature may offer a more targeted and safer intervention. In conclusion, elucidating VA’s role as a microbiome-derived epigenetic regulator would open promising avenues for therapeutic strategies that directly connect gut and CNS health within the GBA.

## 1. Introduction

An increasingly relevant concept in modern medicine is the intricate bidirectional communication between the gut and the brain, known as the gut–brain axis (GBA) [1,2,3]. This complex network is profoundly influenced by the different and dynamic microbial communities residing in the gastrointestinal tract [4,5,6]. This network allows the gut to communicate with the central nervous system (CNS) through multiple channels, including spinal nerves (especially the vagus nerve) and the systemic circulation, which transports microbial-derived molecules and their precursors [7,8,9]. While a state of eubiosis supports host health through the synthesis of vital bioactive compounds, dysbiosis disrupts GBA signaling [10,11,12,13,14,15,16,17,18,19]. Dysbiosis contributes to pathology by compromising the intestinal epithelial barrier, often referred to as leaky gut [20,21]. This hyperpermeability allows microbial products and pro-inflammatory molecules to cross into the bloodstream, triggering chronic, low-grade systemic inflammation [22,23,24]. This, in turn, can compromise the integrity of the blood-brain barrier (BBB), causing the translocation of harmful substances into the CNS and triggering neuroinflammation [25,26].

Neuroinflammation represents a highly coordinated immune response within the CNS, involving the interplay of microglia, astrocytes, neurons, and endothelial cells of the BBB [27,28,29]. Although initially protective, its prolonged state would evolve into a chronic, damaging process characterized by the persistent release of pro-inflammatory cytokines and reactive oxygen species (ROS), leading to neuronal dysfunction and neurodegeneration [30]. Activated microglia [29], often adopting a pro-inflammatory M1 phenotype, release key cytokines such as TNF-α, IL-1β, IL-6, and COX-2, promoting neuronal injury [31]. Concurrently, reactive astrocytes amplify the cascade by secreting VEGF-A, MMP-9, and MCP-1, which increase BBB permeability, alongside IL-6 and oncostatin M that downregulate tight junction proteins such as claudins [32]. This BBB disruption allows the infiltration of peripheral immune cells and the entry of plasma proteins like fibrinogen, which primes microglial NLRP3-inflammasome, driving the sustained release of IL-1β and IL-6 and further inflammatory spread [33,34,35]. Dysbiosis-induced inflammation is a major contributor to this process [36,37]. The progression and severity of chronic neurodegenerative diseases like Alzheimer’s Disease (AD), Parkinson’s Disease (PD), Multiple Sclerosis (MS), and certain forms of Frontotemporal Dementia (FTD) also result from a complex interplay crossing genetic susceptibility variants and environmental factors that shape the epigenome [38,39]. Epigenetic modifications, such as DNA methylation and histone modifications (acetylation, methylation, phosphorylation), and the activity of non-coding RNA (ncRNAs) do not alter the DNA sequence but alter gene expression [40]. This leads to long-lasting modifications crucial to regulatory processes in the CNS, thereby modulating the activity of resident immune cells like microglia and astrocytes [41,42]. For instance, HDAC3 plays a particular pro-inflammatory role, controlling NF-κB activation and the resultant immuno-neuroinflammatory response [43]. Consistently, hypomethylation of NF-κB can lead to the abnormal activation of TNF-α, while histone hyperacetylation can open chromatin and favor the expression of neuroprotective genes, as BDNF [43,44,45,46,47,48,49,50,51,52,53,54,55,56,57,58,59,60,61,62], or as for NLRP3, H3K9-acetylation represents a target for therapeutic modulation [63,64,65].

In this context, among the key players linking the GBA to this epigenetic regulation are Short-Chain Fatty Acids (SCFAs), bioactive molecules produced by gut bacteria through the fermentation of dietary fibers and resistant starches [66]. While the major SCFAs (acetate, propionate, and butyrate) are well known, Valeric Acid (VA)—though less abundant—is emerging as a molecule of significant interest [25]. VA is a microbial-derived SCFA, primarily known for its selective inhibition of Histone Deacetylases (HDACs), along with its neuroprotective and anti-inflammatory properties [67,68].

This review explores the role of VA as an emerging gut-derived mediator in the GBA. It investigates its mechanisms of action, including its influence on neuroinflammatory pathways and neurodegenerative diseases. This analysis provides an updated overview of how this small gut-derived molecule could have a profound impact on brain health and highlights its importance for future research.

## 2. The Gut Microbiota: Eubiosis and Dysbiosis

The gut microbiota refers to the complex and diverse community of bacteria, yeasts, fungi, viruses, and protozoa that predominantly resides in the gastrointestinal tract, harboring over 70% of all microbes within the human body [69]. In recent decades, the application of high-throughput “omics” technologies, such as metagenomics and metabolomics, has been crucial in elucidating the vast complexity of the gut microbiota ecosystem and its interactions with the host [70]. The microbial ecosystem is predominantly composed of five main bacterial phyla: *Firmicutes*, *Bacteroidetes*, *Actinobacteria*, *Fusobacteria*, and *Proteobacteria*, with the addition of *Verrucomicrobia* and *Cyanobacteria*. Among these, *Firmicutes* and *Bacteroidetes* account for over 90% of the total microbial population [71]. Approximately 99% of these microorganisms are anaerobic [4,72].

There is no universally accepted definition of a “healthy” gut microbiota, as it exists in a state of dynamic equilibrium unique to each individual, that ensures resilience against transient external perturbations, such as fluctuations in diet and lifestyle [73,74,75]. This balanced status, called eubiosis, contributes to host health—defined as the absence of disease—and supports normal physiological function. The relationship between the intestinal microbiota and the host is a mutualistic symbiosis, where the microbiota contributes to host homeostasis by synthesizing bioactive compounds such as vitamins, antimicrobial peptides, and SCFAs [76,77]. Conversely, an imbalance in microbial composition that negatively impacts gastrointestinal, immune, or neural function is termed dysbiosis [78]. Dysbiosis is often caused by factors such as an unhealthy diet, chronic stress, and antibiotic use [79,80]. Alterations in dietary habits represent one of the most influential factors shaping the composition of the individual microbial community [81]. Diets characterized by a low intake of microbiota-accessible carbohydrates, particularly soluble and insoluble fibers commonly present in fruits, vegetables, whole grains, legumes, and potatoes, significantly impair the growth of fermentative bacterial taxa, thereby reducing microbial diversity [82]. Such a condition directly compromises the mutualistic symbiosis essential for maintaining host health, giving rise to a stable dysbiotic configuration characterized by an overrepresentation of opportunistic species and an enhanced risk of disease [83,84]. Accordingly, an altered *Firmicutes*/*Bacteroidetes* ratio—specifically, an increase in *Firmicutes* and a decrease in *Bacteroidetes*—is a common feature in several disorders, including those involving inflammatory states as neurodegenerative conditions [85,86].

Clinical manifestations most frequently associated with intestinal dysbiosis include abdominal bloating, excessive gas production, and reduced appetite linked to irregular intestinal motility, together with extraintestinal symptoms such as fatigue, depressive states [87].

This loss of symbiosis directly impacts intestinal mucosal health, which is crucial for nutrient absorption and preventing pathogen passage [88,89]. The effectiveness of this epithelial barrier is controlled by tight junction proteins, including transmembrane proteins (occludins and claudins) and intracellular scaffold proteins (ZO-1, -2, and -3) [90]. A disruption in these proteins leads to leaky gut, allowing inflammatory molecules and microbial products to enter the bloodstream, triggering systemic inflammation. This systemic inflammation, as established, subsequently compromises the integrity of the BBB, whose permeability is similarly controlled by tight junction proteins [91].

## 3. The Role of Short-Chain Fatty Acids (SCFAs): Systemic Impact

SCFAs, including acetate (C2), propionate (C3), and butyrate (C4), typically in a 60:20:20 ratio, alongside lesser amounts of valeric (C5), isobutyric, and isovaleric acids, are produced alongside gases during the fermentation of dietary substrates like fiber and resistant starch—substrates that the host cannot digest due to the lack of a specific repertoire of carbohydrate-active enzymes (CAZymes) [66,92]. Minor SCFAs like isobutyric and isovaleric acids are generated from protein fermentation [90]. Reports consistently show that low-fiber diets are often associated with increased inflammatory disturbances [93], hence adding fibers can counteract negative effects by decreasing inflammatory mediators both in humans and rats [94,95,96,97,98].

The amount and pattern of SCFAs formed are dependent on the type of dietary fiber consumed [99,100]. Butyric acid serves as a primary energy source for colonocytes; in fact, it is promptly metabolized by colonocytes via mitochondrial β-oxidation. By contrast, acetate serves as a substrate for cholesterol biosynthesis, and propionic acid is primarily utilized by hepatocytes for glucose production, which plays a key role in the maintenance of blood glucose levels [66]. Once formed, most SCFAs are rapidly absorbed through the colonic mucosa, but some reach the systemic circulation, where they can have effects on peripheral organs like the liver, heart, and brain (Figure 1) [101]. As weak acids, SCFAs can exist in either protonated or deprotonated form. The protonated SCFAs are easily absorbed via passive diffusion, whereas the deprotonated form requires transporters, including H+-coupled monocarboxylate transporters (MCTs) and sodium-coupled monocarboxylate transporters (SMCTs) [25]. These transporters are expressed in a variety of tissues, including the gut, liver, and brain [101,102]. The regulatory effects of SCFAs on metabolic and immune disorders are mediated by their binding to G-protein coupled receptors (GPRs), such as GPR41 and GPR43 [103]. GPR41 activation is preferred with longer chain lengths (C3–C5), including valeric acid (VA) [104]. Furthermore, SCFAs can regulate the expression of inflammatory mediators by inhibiting histone deacetylases (HDACs), proteins expressed mainly in immune system-mediating cells such as Tregs, antigen-presenting cells (APCs), and effector T cells [105].

## 4. Valeric Acid: Biosynthesis, Metabolism, and Local Effects

Valeric acid (pentanoic acid, VA) is a naturally occurring SCFA with a five-carbon (C5) straight-chain structure, conferring it moderate lipophilicity and efficient membrane permeability [106]. It was named after its origin in the perennial flowering plant *Valeriana officinalis* [107,108], whose extracts have long been used for their sedative effects to treat pain, epilepsy, insomnia, anxiety, and depression [109]. VA is mainly produced by anaerobic gut bacteria through a chain elongation process. One key route involves the condensation of ethanol and propionate (Figure 2). Ethanol is first oxidized to acetyl-CoA (Wood–Ljungdahl pathway), while propionate is activated to propionyl-CoA. These two CoA derivatives undergo thiolase-mediated condensation to form 3-ketopentanoyl-CoA, which is then reduced and converted to VA through CoA-thioester intermediates [110]. This pathway shares similarities with butyrate synthesis and reflects metabolic plasticity based on substrate availability. Notably, the balance of ethanol and propionate in the gut influences whether microbes produce butyrate (C4) or valerate (C5) [110].

Once produced in the colon, VA is absorbed as its ionized form, valerate. Locally, VA contributes to the acidic environment in the colon, which inhibits the growth of pathogenic bacteria (e.g., *E. coli* and *Salmonella*) and promotes beneficial strains (e.g., *Lactobacillus* and *Bifidobacterium*) [111,112]. It is transported by the portal vein to the liver, where it can be metabolized through β-oxidation, and then enters systemic circulation to exert its effects on peripheral organs and the brain [101]. Its physiological role extends beyond energy metabolism, as it acts as a potent signaling molecule. The dietary input, therefore, represents a direct, modifiable link between nutritional status and CNS effects within the GBA [113,114].

## 5. Valeric Acid as an Epigenetic Modulator

VA also exerts epigenetic effects, particularly through inhibition of HDACs, leading to chromatin remodelling and regulation of gene expression [115,116]. HDACs are a family of enzymes crucial for regulating gene expression by removing acetyl groups from histone and non-histone proteins. HDACs are broadly categorized into four main classes based on their sequence homology, cellular localization, and cofactor requirements. Class I (HDAC1, 2, 3, and 8) are primarily nuclear, zinc-dependent enzymes. Class II is divided into Class IIa (HDAC4, 5, 7, and 9), which shuttle between the nucleus and cytoplasm, and Class IIb (HDAC6 and 10), with HDAC6 being largely cytoplasmic; both are also zinc-dependent. Class III comprises sirtuins (SIRT1-7), which are NAD+-dependent. Finally, Class IV contains only HDAC11, sharing features of both Class I and Class II, and is also zinc-dependent. Classes I, II, and IV are often referred to as “classical” HDACs, as their activity is inhibited by trichostatin A (TSA), unlike the Class III sirtuins [117]. VA acts as an epigenetic modulator through its selective inhibition of class I histone deacetylases (HDACs), especially HDAC3 (Figure 3) [116]. Accordingly, Yuille et al. demonstrated that VA produced by commensal *M. massiliensis* promoted a strong HDACi effect in IFN-γ, IL-10, IL-1β, and TNF-α cytokine suppression, contributing to anti-inflammatory effects in ulcerative colitis [116]. Furthermore, Han et al. recently highlighted that VA inhibits the expression of HDAC3 with an IC_50_ of 16.6 µM, therefore suppressing the proliferation of prostate cancer cells [118]. The most likely mechanism of action for VA’s direct inhibition of HDACs is its binding to the zinc ion (Zn^2+^) located in the enzyme’s catalytic site [119]. While this molecular mechanism is compelling, much of the current understanding relies on in vitro cell-based assays with limited data. Therefore, further studies are needed to fully elucidate the specific and dose-dependent mechanisms of VA’s epigenetic modulation within the GBA.

## 6. Valeric Acid: Systemic Effects and Neuroprotection

VA binds to GPR41 and GPR43 in various tissues (liver, skeletal muscle, heart, kidney, pancreas, and brain), modulating key processes such as satiety, insulin sensitivity, and inflammation [101,102,103]. Accordingly, Kumari et al. highlighted that VA inhibited the NF-κB signalling pathway, leading to decreased expression of pro-inflammatory cytokines such as TNF-α and IL-6 [120].

Beyond its local effects, VA also encompasses systemic and neurological implications within the GBA [116,121,122,123]. Direct evidence suggests that VA is a key mediator in the development of Postoperative Cognitive Dysfunction (POCD). The increase of VA in the blood observed after surgery in mice was positively associated with the abundance of *Lactobacillus* and *Anaerotruncus*. This systemic elevation is a likely mechanism for the suppression of brain cell genesis and the compromise of synaptic plasticity, dendritic arborization, and cognitive function. The detrimental effect of VA on learning and memory is potentially mediated by the activation of complement C3 signaling. In particular, while peripheral (intraperitoneal) injection of VA impaired cognition in the model, direct central administration (intracerebroventricular) at a low, brain-relevant dose failed to induce cognitive learning and memory impairments. This discrepancy suggests that the adverse effects of high systemic VA in POCD may be primarily mediated by peripheral inflammatory or complement signaling, rather than a direct, toxic effect of VA on CNS neurons at concentrations typically achieved in the brain. Crucially, low-intensity exercise has been shown to stabilize the gut microbiota against surgical insult and mitigate POCD development. These results suggest that reducing blood VA by altering the gut microbiota may represent a primary mechanism for the exercise-induced protection against both the immune/neuroinflammatory responses and the suppression of brain cell genesis. In this context, VA is identified as a potential therapeutic target for brain health in clinical settings involving major surgery [122]. Consistently, a higher level of VA produced by *Actinomycetaceae* has also been reported in children with autism spectrum disorder (ASD). In particular, a range of VA concentrations among children with ASD was observed, suggesting high inter-individual variability and the possibility of different endophenotypes. Furthermore, intermediate levels of VA in unaffected siblings suggest a contribution from common genetic or environmental factors. This finding suggests that VA in ASD is likely multifactorial, and its physiological functions in regulating the GBA remain under investigation [124].

A key modulatory effect of VA is on the GABAergic system [125]. VA structurally resembles gamma-aminobutyric acid (GABA), the primary inhibitory neurotransmitter in the CNS, and has been shown to bind the GABA_A receptor. This raises the possibility that bacteria involved in VA production and metabolism could be associated with depression, bridging the gap between microbial metabolites and psychiatric symptoms [123]. Anuszkiewicz et al. demonstrated that VA can influence cerebral vascular tone and neuroinflammation through colon-vagus nerve signalling mediated by GPR41/43, unveiling new therapeutic avenues for neurovascular disorders [126]. Foundational pharmacological characterizations of GABA_A receptors by Edvinsson and Krause [127] and Lloyd and Dreksler [128] further affirmed their central role in cerebral vascular regulation and neuronal excitability, mechanisms likely underlying the anxiolytic and neuroprotective effects attributed to VA. Vishwakarma et al. further reported that VA exerts a protective effect against cognitive impairment induced by intracerebroventricular streptozotocin (ICV-STZ) in rats, suggesting a possible GABAergic mechanism underlying its neuroprotective action. In this dementia model, VA improved cognitive performance, likely by enhancing GABAergic transmission and reducing cerebral oxidative stress [109]. Similarly, Nouri and Abbas Abad observed that the aqueous extract of *Valeriana officinalis* root increases the threshold for picrotoxin (PTZ)-induced clonic seizures in mice, further supporting the involvement of the GABA system as a pharmacological target of VA [129]. These findings suggest that VA acts as a positive modulator of GABA_A receptors, contributing both to the stabilization of neuronal activity and to the potential prevention of neurodegenerative and convulsive disorders. Moreover, Mulyawan et al. reported increased expression of the GABRB3 subunit of the GABA_A receptor following administration of valerian extracts in BALB/c mice, strengthening the evidence for neuroprotective modulation [130].

Taken together, this body of evidence suggests that VA acts not just as a microbial metabolite with local effects but as a bioactive compound with the capacity to systemically modulate neuroprotective effects on host epigenetics. Despite the compelling preclinical findings, its translatability warrants further investigation into strategies that leverage gut-derived interventions to counteract inflammation, cancer, and metabolic disorders [120].

## 7. From a Pharmacological Drug to a Physiological Mediator: Valproic Acid and Valeric Acid in Epigenetics

Both Valeric Acid (VA) and its pharmacological derivative, Valproic Acid (VPA), have garnered increasing interest due to their shared ability to act as Histone Deacetylase inhibitors (HDACi) [131]. While VA selectively targets Class I HDACs, VPA acts as a well-known, non-selective HDACi, targeting multiple enzyme classes [132,133]. VPA primary targets include Class I HDACs and Class IIa HDACs [117,134,135] with a mechanism that involves the chelation of zinc ions (Zn^2+^) located within the catalytic site. This disruption leads to an accumulation of acetyl groups on lysine residues within histones, H3K9ac, H3K14ac, and H4K5ac [135]. Beyond directly influencing histone acetylation, VPA can also indirectly alter DNA methylation patterns or interact with chromatin remodeling complexes (SWI/SNF family), significantly broadening its epigenetic impact and strengthening its ability to orchestrate complex changes across the epigenome [136,137]. This fundamental epigenetic mechanism underlies VPA’s neuroprotective and anti-inflammatory effects. VPA activity on the epigenome is particularly relevant in diseases, where epigenetic dysregulations act as both consequences and drivers of pathogenesis (Table 1). Despite preclinical evidence supporting the neuroprotective and anti-inflammatory potential of VA and VPA, translating these findings into clinical practice for neuroinflammatory and neurodegenerative diseases remains complex.

Unlike VPA, which is administered as a pharmaceutical drug acting as a broad-spectrum pharmaceutical derivative, VA acts locally in the gut by modulating the microbiota and then enters systemic circulation to exert its effects, which also include HDAC inhibition at the brain level. This function stems from its intestinal origin. In this sense, VA functions as a more integrated and physiological “epigenetic mediator,” defined here as an endogenously produced metabolite, derived from the host microbiota, which exerts its effects by natural homeostatic mechanisms rather than pharmacological intervention. Indeed, the primary advantage of VA selectively targeting the inhibition of HDAC3 (compared to the non-selective VPA) lies in a better safety profile and greater therapeutic specificity, thereby reducing harmful off-target side effects (Table 2). Moreover, VA’s physiological nature conflicts with the need for a sustained, pharmacologically relevant concentration necessary to maintain an epigenetic effect in the CNS over time. Thus, the potential contribution of VA lies in disease management through nutritional or microbial modulation aimed at optimizing VA levels and maintaining CNS homeostasis within the host physiology and the systemic toxicity of VPA [170,194,195]. Recent studies highlighted the direct epigenetic effects of VA. Jayaraj et al. demonstrated in neurodegeneration models that VA protects dopaminergic neurons by reducing oxidative stress, neuroinflammation, and modulating autophagy pathways—pointing to direct epigenetic mechanisms with potential relevance for Parkinson’s disease (PD) and other neurodegenerative conditions [170]. This evidence supports the therapeutic potential of both VA and VPA in managing diseases where epigenetic dysregulations drive pathogenesis. Despite the compelling preclinical advantages of VA’s selectivity over VPA’s broad action, it is essential to note that the clinical translation of VA’s epigenetic potential is still in its nascent stage.

## 8. Clinical and Epigenetic Challenges in VA Research

The current stage of VA research is marked by limitations that present unique opportunities for future study. The primary challenge is the absence of dedicated clinical trials on pure VA and its derivatives as epigenetic therapeutics. Most existing data are derived indirectly from studies on its synthetic precursor, VPA, despite significant differences in their pharmacokinetic profiles and HDAC selectivity. Moreover, the absence of validated epigenetic and neuroinflammatory biomarkers in human biofluids remains a major hindrance. However, the major challenge concerns the proper concentration required to engage the epigenetic effect compared to the physiological circulating levels of VA. The circulating level of VA in human plasma is 0.18 µM [118], which is substantially lower than the concentration used in in vitro studies. Furthermore, this discrepancy is further highlighted by the fact that in vitro models do not fully recapitulate the complexities of in vivo pharmacokinetics, compartmentalization, target engagement kinetics, and metabolism in the CNS microenvironment. Therefore, further studies are needed to determine the effective VA concentration necessary to achieve HDAC3 inhibition in vivo at the CNS level. While the literature suggests that VA possesses the capacity to cross the BBB, the critical gap remains the lack of data confirming that VA achieves concentrations sufficient to engage its target in vivo [25,196]. Furthermore, the current literature lacks studies that fully map the specific epigenetic signature and downstream transcriptional changes following VA-induced HDAC3 inhibition in neurons and glia. To bridge the gap between preclinical data and clinical application, researchers should prioritize specific VA-focused studies. This would include utilizing advanced in vitro models, such as those employing induced pluripotent stem cells (iPSCs) derived from patients [197,198,199,200]. These models offer a genetically human context to test the compound and quantify its selective HDAC inhibition on patient-specific cellular mechanisms, providing a crucial intermediate step before large-scale human trials.

Another major practical hurdle is VA’s inherent pharmacological profile. As an endogenously produced SCFA, VA suffers from rapid systemic clearance and substantial first-pass metabolism in the gut and liver [93,99,201]. Achieving therapeutic effectiveness in the CNS would require dedicated pharmacological strategies, such as prodrug development or targeted delivery systems. Furthermore, like other endogenously produced molecules, VA faces regulatory challenges regarding its classification (drug vs. dietary supplement) and requires robust, controlled clinical data to definitively prove its efficacy and safety profile for clinical application.

The critical necessity to overcome these challenges is underlined by the severe limitations of VPA. The drug’s safety profile and significant side effects pose a major barrier to its broader application. These adverse effects at therapeutic dosages include hepatotoxicity (with risks of acute liver failure), teratogenicity (e.g., neural tube defects if taken during pregnancy), common gastrointestinal effects (nausea, vomiting), neurological effects (sedation, tremor, cognitive slowing), hematological effects (thrombocytopenia), and the potential for weight gain and features of metabolic syndrome [170,194,195,202]. These considerable risks substantially limit VPA’s chronic use, especially in vulnerable populations or when clear benefits do not demonstrably outweigh the risks in complex neurodegenerative pathologies.

## 9. Future Perspectives

In conclusion, although preclinical data suggest valeric acid (VA) could be used as a naturally derived, bioactive compound for neuroprotection, the successful and safe translation of these findings into clinical practice warrants multidisciplinary efforts. This includes the development of innovative pharmacological strategies alongside thorough and methodologically rigorous clinical trials to validate its efficacy and safety profile. Should these efforts yield positive results, VA could emerge as an alternative conceptual approach to valproic acid (VPA) by leveraging selective, host-derived metabolites to address the complex pathophysiology of central nervous system disorders.

## Figures and Tables

**Figure 1 cells-14-01823-f001:**
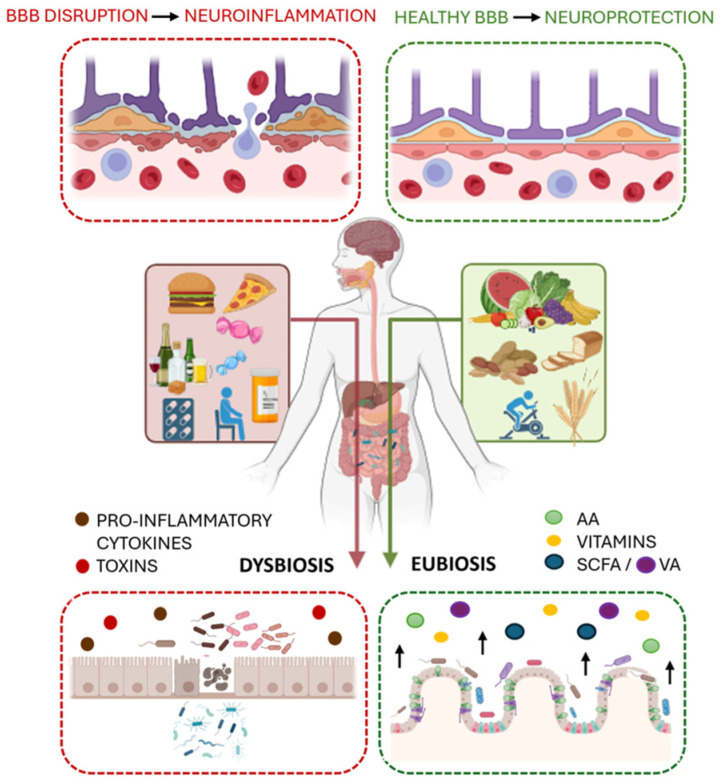
The gut–brain axis. The figure illustrates how a healthy gut microbiota (eubiosis) produces beneficial metabolites like short-chain fatty acids (SCFAs), among which valeric acid (VA) to support gut and brain health, while an imbalanced microbiota (dysbiosis) leads to inflammation and neuroinflammation. This figure was created in BioRender.com. (C.P.). (2025) https://www.biorender.com/.

**Figure 2 cells-14-01823-f002:**
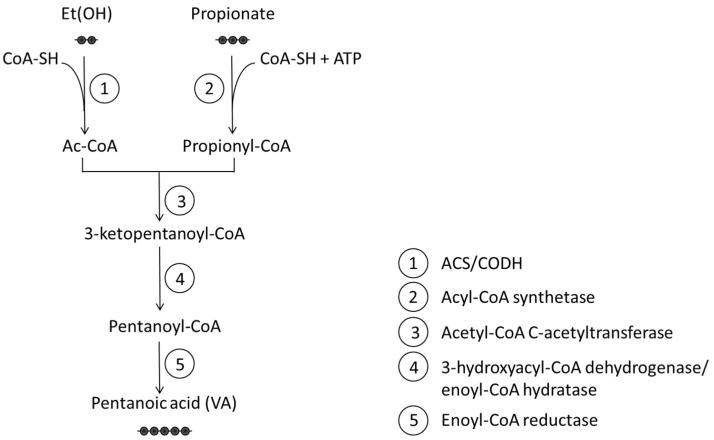
Schematic representation of Valeric Acid (VA) biosynthesis pathway. Ethanol (C2) and propionate (C3) are condensed to form pentanoic acid (VA, C5) in the gut microbiota. Enzymatic steps: 1: Acetyl-CoA Synthase/Carbon Monoxide Dehydrogenase (ACS/CODH); 2: Acyl-CoA synthetase; 3: Acetyl-CoA C-acetyltransferase; 4: Intermediate oxidoreductases (3-hydroxyacyl-CoA dehydrogenase and enoyl-CoA hydratase); 5: Enoyl-CoA reductase. Et(OH): Ethanol; CoA-SH: Coenzyme A; Ac-CoA: acetyl-CoA; ATP: Adenosine triphosphate.

**Figure 3 cells-14-01823-f003:**
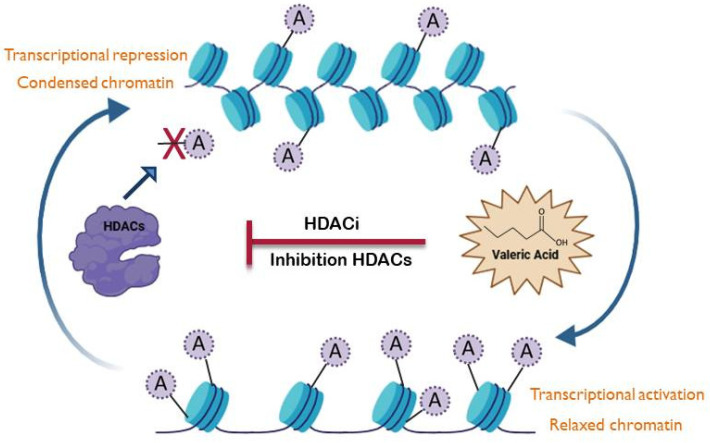
Valeric acid as an epigenetic modulator. The figure illustrates how valeric acid (VA) acts as a Histone Deacetylase Inhibitor (HDACi), promoting the acetylation of histone proteins. This preservation of histone acetylation leads to chromatin decondensation and consequent modulation of inflammation. This figure was created in BioRender.com. (C.P.). (2025) https://www.biorender.com/.

**Table 1 cells-14-01823-t001:** Valproic Acid (VPA) and Valeric Acid (VA) in neurological disorders. Comparative overview of preclinical and clinical evidence for VPA and VA in the epigenetics of neurological disorders.

Disorder	Genetic Cause	Epigenetic Mechanism & Rationale	Preclinical/Clinical Evidence (VPA)	Valeric Acid (VA)	References
Rett Syndrome	*MECP2* mutations	VPA, as an HDACi, is theoretically relevant to counteract MECP2 dysfunction.	Limited clinical evidence. Used for seizure management, not a disease-modifying therapy due to severe adverse effects.	N/A	[138,139,140,141]
Aicardi-Goutières Syndrome (AGS)	Mutations in genes like *TREX1*, *SAMHD1*.	Chronic activation of the cGAS-STING pathway and Type I interferon production leads to neuroinflammation. VPA might downregulate IFN-stimulated genes (ISGs) and pro-inflammatory cytokines.	Primarily preclinical research. Epigenetic modulation remains a theoretical approach.	N/A	[142,143,144,145,146,147,148,149,150]
Autism Spectrum Disorder (ASD)	Multifactorial, including de novo mutations (e.g., *ADNP*, *MECP2*, *SHANK3*) and polygenic risk.	Environmental factors and genetic variants shape the epigenome. VPA’s epigenetic action and impact on gut microbiota could modulate ASD-like behaviors.	Mixed clinical evidence. Prenatal VPA exposure is a known risk factor for ASD. Therapeutic use in established ASD is limited, though some small trials have shown a reduction in repetitive behaviors.	VA as a potential pathological mediator/marker.	[121,151,152,153,154,155,156,157,158,159,160]
Genetic FTD, AD, PD	Mutations in *GRN*, *C9orf72* (FTD), *APP*, *PSEN1* (AD), *LRRK2* (PD).	Epigenetic dysregulations contribute to pathogenesis, including protein aggregation (TDP-43, alpha-synuclein), mitochondrial dysfunction, and oxidative stress. VPA could shift glial cells from a pro-inflammatory to a neuroprotective phenotype.	Variable preclinical results. Use in human studies is restricted by safety concerns, but preclinical models have explored its potential.	Protection of dopaminergic neurons by reducing oxidative stress and modulating autophagy.	[161,162,163,164,165,166,167,168,169,170]
Amyotrophic Lateral Sclerosis (ALS)	Mutations in *SOD1*, *C9orf72*, *FUS*, *TDP-43*.	Epigenetic dysregulation, particularly involving HDACs. HDAC2 levels are increased in some ALS samples. However, the role of specific HDACs like HDAC4 is complex and context-dependent.	Complexity in preclinical findings. Studies on HDAC4 inhibition in mouse models have shown detrimental effects, highlighting the nuanced and sometimes contradictory role of individual HDAC isoforms.	N/A	[171,172,173,174,175,176]
Spinal Muscular Atrophy (SMA)	Mutations in *SMN1/2* genes.	VPA, as an HDACi, can enhance SMN2 expression at RNA and protein levels. This effect is more pronounced in patients with a higher SMN2 copy number.	Mixed clinical results as a monotherapy. VPA was shown to increase SMN levels but provided limited benefits for motor deficits. Recent studies suggest VPA may act synergistically with other treatments like nusinersen.	N/A	[177,178,179,180,181,182,183,184,185]
Multiple Sclerosis (MS)	Multifactorial; genetic variants in *HLA-DRB1* and other immune genes.	Genetic and environmental factors shape the epigenome. VPA modulates immune responses, promotes oligodendrocyte differentiation, and increases myelination gene expression.	Promising preclinical evidence in EAE models, showing reduced disease severity, inflammation, and demyelination. Clinical research is ongoing, but therapeutic use is not yet standardized due to safety concerns.	N/A	[186,187,188,189,190,191,192,193]

Abbreviation: VA: Valeric acid; VPA: Valproic acid; N/A: not available; EAE: Experimental Autoimmune Encephalomyelitis; MECP2: Methyl-CpG-Binding Protein 2; ADNP: Activity-Dependent Neuroprotector Homeobox; SHANK3: SH3 and Multiple Ankyrin Repeat Domains 3; TREX1: 3′-Repair Exonuclease 1; SAMHD1: SAM and HD Domain Containing dNTP Hydrolase 1; GRN: Granulin Precursor Gene; C9orf72: Chromosome 9 Open Reading Frame 72; APP: Amyloid Precursor Protein; PSEN1: Presenilin 1; LRRK2: Leucine-Rich Repeat Kinase 2; SOD1: Superoxide Dismutase 1; FUS: Fused in Sarcoma; TDP-43: TAR DNA-Binding Protein 43; SMN1/2: Survival Motor Neuron 1/2; HLA-DRB1: Human Leukocyte Antigen-DR Beta 1.

**Table 2 cells-14-01823-t002:** Similarities and differences between VA and VPA. A comparative analysis of clinical attributes that differentiate the endogenous VA from the pharmaceutical drug VPA.

Comparative Features	Valeric Acid (VA)	Valproic Acid (VPA)
Origin	Gut Microbiota (Endogenous Metabolite)	Synthetic Drug (Pharmaceutical)
HDAC Targets	Selective, primarily Class I (HDAC3)	Non-selective, Class I & Class IIa
Effective Concentration (EC_50_)	Physiological Circulating levels = 0.18 µM); HDAC3 IC_50_ (in vitro) = 16.6 µM	50–125 μg/mL (mM range)
Toxicity/Side Effects	Low/Minimal (Physiological)	High/Severe (Hepatotoxicity, Teratogenicity)
Primary Role	Physiological Epigenetic Mediator	Established Anticonvulsant/Mood Stabilizer
Brain Bioavailability	Effective permeability	Effective permeability
Clinical Status	Preclinical/Early-stage research	Established Drug

## Data Availability

Not applicable.

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
