# Peer review of "Valeric Acid: A Gut-Derived Metabolite as a Potential Epigenetic Modulator of Neuroinflammation in the Gut–Brain Axis"

_cells, 2025, doi:10.3390/cells14221823_

Round 1
Reviewer 1 Report
Comments and Suggestions for Authors
This review manuscript addresses a timely and significant topic: the role of the gut-derived metabolite valeric acid (VA) as an epigenetic modulator in the context of the gut-brain axis and neuroinflammation. The article is well-structured and provides a useful overview of the topic. However, a more critical and balanced perspective is required before it can be considered for publication. The current version overstates the therapeutic potential of VA while overlooking significant gaps in evidence and contradictory findings.
- The authors should adopt a more critical tone throughout the manuscript. This includes explicitly discussing the limitations of cited studies (e.g., in vitro vs. in vivo, physiological concentrations, model systems), addressing contradictory evidence, and clearly distinguishing between established facts and speculative hypotheses. Language should be moderated to reflect the current state of evidence (e.g., replace promise and potential with more cautious terms like warrants further investigation).
- The core argument that VA exerts neuroprotection via selective inhibition of HDAC3 is not adequately substantiated.The manuscript fails to provide direct evidence of VA crossing the blood-brain barrier (BBB) at physiologically relevant concentrations to engage with its target in the CNS. The cited IC50 value for HDAC3 inhibition (16.6 µM) is significantly higher than the typical circulating levels of VA (<5 µM).
- The mechanistic chain (HDAC3 inhibition → specific gene expression changes → neuroprotection) is incomplete. The review needs to discuss which specific genes are targeted and how their altered expression translates into functional outcomes in neurons or glia.
- The manuscript mentions studies where elevated VA levels are associated with negative outcomes (e.g., in Postoperative Cognitive Dysfunction and Autism Spectrum Disorder) but fails to critically analyze the implications of these findings.
- While the comparison to VPA is a good framing device, the analysis in Section 7 and Table 2 is overly simplistic. The argument that selective HDAC3 inhibition is inherently superior to broad-spectrum HDAC inhibition is not critically evaluated.
- Restructure the manuscript to introduce the epigenetic hypothesis earlier, perhaps immediately after the general introduction to SCFAs.
- Section 8 Clinical and epigenetic challenges in VA research is underdeveloped and lacks depth. It briefly mentions key hurdles without substantive discussion.
- Certain phrases are overly optimistic (e.g., new therapeutic paradigm). The term physiological epigenetic mediator is novel and requires a clear, operational definition.
Author Response
This review manuscript addresses a timely and significant topic: the role of the gut-derived metabolite valeric acid (VA) as an epigenetic modulator in the context of the gut-brain axis and neuroinflammation. The article is well-structured and provides a useful overview of the topic. However, a more critical and balanced perspective is required before it can be considered for publication. The current version overstates the therapeutic potential of VA while overlooking significant gaps in evidence and contradictory findings.
We thank the reviewer in taking time to revise our manuscript and for his insightful comments. We improved the manuscript as follows:
- The authors should adopt a more critical tone throughout the manuscript. This includes explicitly discussing the limitations of cited studies (e.g., in vitro vs. in vivo, physiological concentrations, model systems), addressing contradictory evidence, and clearly distinguishing between established facts and speculative hypotheses. Language should be moderated to reflect the current state of evidence (e.g., replace promise and potential with more cautious terms like warrants further investigation).
We improved Section 5 to include a more nuanced findings from the route of administration (systemic vs. central) of VA in the POCD. We also improved the findings relative to ASD and the inter-individual variability. We also concluded in Section 6 that although neuroprotective and GABAergic effects are promising, taken together the preclinical findings, the translability is limited.
We improved all sections with a more moderate evidence-based tone.
- The core argument that VA exerts neuroprotection via selective inhibition of HDAC3 is not adequately substantiated. The manuscript fails to provide direct evidence of VA crossing the blood-brain barrier (BBB) at physiologically relevant concentrations to engage with its target in the CNS. The cited IC50 value for HDAC3 inhibition (16.6 µM) is significantly higher than the typical circulating levels of VA (<5 µM).
We agree that the direct evidence required to validate this hypothesis is currently the biggest limitation in the field. To address this comprehensively we improved the discussion in Section 8. The IC50 determined in in vitro models does not account for the complexities of in vivo pharmacokinetics and pharmacodynamics (DOI: 10.1007/s12032-022-01814-9). Therefore further studies are needed. Moreover, regarding the BBB permeability, we clarified that while literature suggests VA possesses the capability to cross the BBB (DOI: 10.3389/fendo.2020.00025; DOI: 10.1038/ncomms4611), the critical gap is not the crossing itself, but the lack of data confirming that VA achieves effective concentration to be biologically active.
- The mechanistic chain (HDAC3 inhibition → specific gene expression changes → neuroprotection) is incomplete. The review needs to discuss which specific genes are targeted and how their altered expression translates into functional outcomes in neurons or glia.
We fully agree that the complete mechanistic chain is currently incomplete and represent a knowledge gap in VA research. We remarked this point in Section 8.
- The manuscript mentions studies where elevated VA levels are associated with negative outcomes (e.g., in Postoperative Cognitive Dysfunction and Autism Spectrum Disorder) but fails to critically analyze the implications of these findings.
We improved former section 5, actual section 6, to integrate a critical analysis in POCD and ASD.
- While the comparison to VPA is a good framing device, the analysis in Section 7 and Table 2 is overly simplistic. The argument that selective HDAC3 inhibition is inherently superior to broad-spectrum HDAC inhibition is not critically evaluated.
We improved the following changes to address accuracy and critical discussion. The Effective Concentration entry in Table2 has been improved highlighting the proper concentration. We clarified that VA's potential lies in being a preventative homeostatic mediator that could be modulated through diet for prevention, distinguishing the VA strategy from VPA.
- Restructure the manuscript to introduce the epigenetic hypothesis earlier, perhaps immediately after the general introduction to SCFAs.
We agree that this change significantly improves the clarity and narrative flow of our central argument. We have restructured the manuscript by establishing the mechanistic link of VA actions following the suggestion of the reviewer.
- Section 8 Clinical and epigenetic challenges in VA research is underdeveloped and lacks depth. It briefly mentions key hurdles without substantive discussion.
We implemented Section 8 in order to consider the key pharmacological and mechanistic gaps.
- Certain phrases are overly optimistic (e.g., new therapeutic paradigm). The term physiological epigenetic mediator is novel and requires a clear, operational definition.
We thank the Reviewer for this insightful and constructive comment. We added a better explanation of epigenetic mediator in our context and revised the terminology properly. In section 7 the corrections are the following: A physiological epigenetic mediator is defined here as an endogenously produced metabolite, derived from the host microbiota, which exerts its effects by natural homeostatic mechanisms rather than pharmacological intervention. [...] Thus, the potential contribution of VA lies in disease management through nutritional or microbial modulation aimed at optimizing VA levels to maintain CNS homeostasis within the host physiology without the systemic toxicity of VPA [...].
We also revised Section 9 to adopt a more appropriate tone.
Reviewer 2 Report
Comments and Suggestions for Authors
This well-written review provides a clear and comprehensive synthesis of current knowledge on how the gut microbiome and its metabolite valeric acid (VA) influence neuroinflammation and neurological disorders. By integrating microbial metabolism, epigenetic regulation, and immune signaling, the article effectively highlights VA’s therapeutic potential and offers valuable insight into future directions for microbiome-based interventions.
I wish there were a little more about the relationship between diet-gut microbiome-VA-neuroinflammation, but it's not critical.
Author Response
This well-written review provides a clear and comprehensive synthesis of current knowledge on how the gut microbiome and its metabolite valeric acid (VA) influence neuroinflammation and neurological disorders. By integrating microbial metabolism, epigenetic regulation, and immune signaling, the article effectively highlights VA’s therapeutic potential and offers valuable insight into future directions for microbiome-based interventions.
- I wish there were a little more about the relationship between diet-gut microbiome-VA-neuroinflammation, but it's not critical.
We thank Reviewer 2 for his constructive comment. We agree that diet is an initial and crucial element of the Gut-Brain Axis (GBA). We improved the statement in Section 4, along with additional references for a comprehensive review: The dietary input, therefore, represents a direct, modifiable link between nutritional status and CNS effects within the GBA [10.3390/nu14163250; 10.3390/nu12061654].
Reviewer 3 Report
Comments and Suggestions for Authors
Avoid the use of "we" and "our" in the manuscript.
Be consistent with abbreviations in the whole document. You need to define all abbreviations in the manuscript first, then use the abbreviated version.
Avoid starting sentences with an abbreviation.
The introduction is very informative, I suggest reorganizing the paragraphs in the introduction.
I suggest reorganizing the paragraphs in the whole document.
L121: add reference
L145: there are more enzyme, revise
L180-187: I suggest using a graph
Abbreviations: list all abbreviations that were used in the paper.
Author Response
We thank the reviewer for his precious comments. We improved the manuscript as follows:
- Avoid the use of "we" and "our" in the manuscript.
The abstract has been modified according to the suggestion of the reviewer. Line 19-21: This review discusses […]; The analysis compares […].
- Be consistent with abbreviations in the whole document. You need to define all abbreviations in the manuscript first, then use the abbreviated version. Avoid starting sentences with an abbreviation.
The abbreviations were revised consistently in the entire manuscript in accordance to the suggestion of the reviewer.
- The introduction is very informative, I suggest reorganizing the paragraphs in the introduction.
We thank the reviewer for his constructive comment. We improved the introduction avoiding a too deep molecular detail in favor of a more focused functional one in respect to VA.
- I suggest reorganizing the paragraphs in the whole document.
We agree that this change significantly improves the clarity and narrative flow of our central argument. We have restructured the manuscript by establishing the mechanistic link of VA actions following the suggestion of the reviewer.
- L121: add reference
We added the following reference: DOI:10.3390/microorganisms10122507.
- L145: there are more enzyme, revise
We modified Line 145 to indicate that a plurality of host-encoded enzymes is missing meeting the suggestion of the reviewer.
- L180-187: I suggest using a graph
We fully agree with the Reviewer's suggestion. We have included a new figure 2 with the relative captation. Figure 1 is unchanged, while former Figure 2 is now renamed Figure 3.
- Abbreviations: list all abbreviations that were used in the paper.
We revised the full list of abbreviations in accordance to the suggestion of the reviewer.
Round 2
Reviewer 1 Report
Comments and Suggestions for Authors
Good to go